# Sustained elevation of soluble B- and T-lymphocyte attenuator predicts long-term mortality in patients with bacteremia and sepsis

**Anna Lange**[1]*, **Sara Cajander**[1,2], **Anders Magnuson**[3], **Kristoffer Strålin**[4,5], **Olof Hultgren**[6]

1 Department of Infectious Diseases, Faculty of Medicine and Health, Örebro University, Örebro, Sweden, 2 School of Medical Sciences, Faculty of Medicine and Health, Örebro University, Örebro, Sweden, 3 Clinical Epidemiology and Biostatistics, School of Medical Sciences, Örebro University, Örebro, Sweden, 4 Department of Infectious Diseases, Karolinska University Hospital, Stockholm, Sweden, 5 Department of Medicine Huddinge, Karolinska Institutet, Stockholm, Sweden, 6 Department of Clinical Immunology and Transfusion Medicine, Faculty of Medicine and Health, Örebro University, Örebro, Sweden

* anna.lange@regionorebrolan.se

**Data Availability Statement:** All relevant data are within the paper and its Supporting Information files.

## Abstract

Soluble B and T lymphocyte attenuator (sBTLA) has been shown to be associated with severity and outcome, in critically ill septic patients. We aimed to assess the dynamic expression of sBTLA, as a prognostic biomarker of long-term mortality in patients with bloodstream infection (BSI) and sepsis, and to evaluate its association with biomarkers indicative of inflammation and immune dysregulation. Secondarily, sBTLA was evaluated in association with severity and bacterial etiology. Patients with BSI (n = 108) were prospectively included, and serially sampled from admission to day 28. Blood and plasma donors (n = 31), sampled twice 28 days apart, served as controls. sBTLA concentration in plasma was determined with enzyme-linked immunosorbent assay. Associations between sBTLA on day 1–2 and 7, and mortality at 90 days and 1 year, were determined with unadjusted, and adjusted Cox regression. Differences related to severity was assessed with linear regression. Mixed model was used to assess sBTLA dynamics over time, and sBTLA associations with bacterial etiology and other biomarkers. sBTLA on day 1–2 and 7 was associated with mortality, in particular failure to normalize sBTLA by day 7 was associated with an increased risk of death before day 90, adjusted HR 17 (95% CI 1.8–160), and one year, adjusted HR 15 (95% CI 2.8–76). sBTLA was positively associated with CRP, and negatively with lymphocyte count. sBTLA on day 1–2 was not linearly associated with baseline SOFA score increase. High SOFA (≥4) was however associated with higher mean sBTLA than SOFA ≤3. sBTLA was not associated with bacterial etiology. We show that sustained elevation of sBTLA one week after hospital admission is associated with late mortality in patients with BSI and sepsis, and that sBTLA concentration is associated with CRP and decreased lymphocyte count. This suggests that sBTLA might be an indicator of sustained immune-dysregulation, and a prognostic tool in sepsis.

**Funding:** The study was supported by grants from Region Örebro Län https://nyckelfonden.regionorebrolan.se/sv and https://www.researchweb.org/is/oll, (AL, SC, KS), Stiftelsen Olle Engkvist Byggmästare https://engkviststiftelserna.se (OH), and Signe och Olof Wallenius stiftelse https://www.signeolofwalleniusstiftelse.se (OH). The funders had no role in study design, data collection and analysis, decision to publish, or preparation of the manuscript.

**Competing interests:** The authors have declared that no competing interests exist.

## Introduction

There is growing interest in long-term sepsis outcomes. While pre-existing comorbidities contribute to late deaths, epidemiological studies indicate that sepsis alone is associated with increased mortality in the subsequent months to years [1, 2].

Evidence suggests that a dysregulated immune response, with persistent inflammation and immunosuppression, may increase the risk of late complications and mortality in sepsis [3–5]. In sepsis survivors, persistent inflammation and immunosuppression can be detected as long as one year after the original sepsis episode [5, 6]. Advances in the understanding of the sepsis immune response has contributed to the new international definition of sepsis, and motivated interventional trials aiming to restore immune homeostasis, but much remains to be outlined in terms of the mechanisms behind immune dysregulation, and how and when it should be assessed [7–10].

Checkpoint inhibitor immunotherapy has transformed cancer treatment, by amplifying ant-tumor responses, and has been found to trigger immune-related adverse reactions, such as colitis and skin rash [11, 12]. PD-1, CTLA-4, and their ligands, have been shown to be upregulated on T cells during sepsis, which has led to investigations on immune checkpoint inhibition also as immunomodulatory treatment in infections [13, 14]. Pre-clinical studies provide evidence that blockade of PD-1 and CTLA-4 could reverse immune dysfunction and improve survival in sepsis, and three recently published phase-1 studies showed that PD-1 and PD-L1 antibody administered to patients with severe sepsis could restore immune cell function in septic patients with low lymphocyte count [9, 10, 15].

B and T lymphocyte attenuator (BTLA) is an important membrane-associated immune-regulatory protein, which is expressed on antigen-presenting cells, T and B lymphocytes and innate immune cells, and is up-regulated after T cell activation [16]. It shares its counter-receptor, herpes virus entry mediator (HVEM), which is also widely expressed by immune cells (T cells, B cells, NK cells, DC and myeloid cells), with other regulatory proteins, and is therefore considered a more fine-tuned immune modulator than PD-1 and CTLA-4 [17]. The BTLA-HVEM pathway has gained recent interest in cancer immunotherapy and autoimmunity [18, 19]. Interestingly, an anti-BTLA-antibody (TAB004) was approved for clinical trials in 2019, and is currently under investigation for treatment of advanced malignancy (NCT04137900).

Co-stimulatory molecules exist in both membrane-bound and soluble forms, which lack the transmembrane domain, and are produced either by shedding of the membrane form, or through alternative mRNA splicing [20]. Transcripts coding for soluble BTLA has been identified in B cells, CD4 and CD8 T cells [21]. We have previously shown that soluble BTLA (sBTLA) is associated with disease severity and short-term mortality, in severe sepsis and septic shock in the intensive care unit (ICU) setting, and others have further demonstrated that sBTLA can discriminate sepsis from other critical illness [22, 23]. Furthermore it was recently shown that sBTLA is associated with an increased risk of death in cancer patients [24–26].

In this study of patients with community-onset BSI, we set out to establish whether sBTLA is associated with mortality at 90 days and one year, and with markers of inflammation and immunosuppression over time. Additionally, we aimed to study sBTLA dynamics over 4 weeks in BSI compared to healthy controls. Finally, we wanted to address whether sBTLA is associated with disease severity, and whether there is a difference in sBTLA production with respect to bacterial etiology.

We found that sustained elevation of sBTLA one week after hospital admission is associated with late mortality in patients with BSI and sepsis, and that sBTLA concentration is associated with decreased lymphocyte count.

## Materials and methods

### Setting, study population and definitions

A prospective single-center observational study of patients with community-onset blood-stream infection was conducted at Örebro University Hospital, Örebro, Sweden, between November 2010 and May 2014. Patients >18 years of age, admitted to the Departments of Infectious Diseases and Internal Medicine with a suspected infection, and in whom a blood culture drawn on hospital admission (day 0) showed growth of clinically significant bacteria within 3 days after hospital admission, were eligible for inclusion, and asked for study participation by a medical doctor. Exclusion criteria were infection with HIV, Hepatitis B and C or previous inclusion in the study. The prevalence of the diagnoses in exclusion criteria are low in the studied population, therefore the sample was considered to be representative of the studied population. The study was approved by the regional ethical review board in Uppsala, Sweden. A written informed consent was obtained from patients or, if unable to sign, a relative.

The increase from each patient's baseline Sequential Organ Failure Assessment (SOFA) score was calculated for the first 24 hours after hospital admission, and is referred to as day 1 Δ-SOFA. Blood samples were drawn from patients on day(s) 0, 1–2 (day of enrolment), 3, 7±1, 14±2, and 28±4 after admission, for biochemical tests, and analyzed with accredited routine laboratory methods. Blood was also collected for processing of plasma for future analyses. Monocytic HLA-DR (mHLA-DR) expression on monocytes was measured from day 1–2 with flow cytometry. HLA-DR data is described in detail elsewhere [27]. Patient data and information on outcome was abstracted from medical records. Healthy blood and plasma donors (n = 31) served as control subjects and were sampled twice, 4 weeks apart. Plasma was kept at -80˚C pending analysis.

### Blood and other cultures

Two sets of blood cultures, each consisting of 20 ml blood distributed equally between one aerobic and one anaerobic bottle, were incubated in a Bactec blood culturing system (Becton Dickinson, Franklin Lakes, NJ, USA). The bacterial species was determined by routine laboratory diagnostic procedures. Other cultures (urine, expectorated sputum or nasopharyngeal aspirate, cerebrospinal fluid, wound swab, joint aspirate) were taken depending on clinical suspicion of diagnosis, according to clinical routine.

### Definitions

Sepsis was defined as an increase of the SOFA score of 2 points or more from baseline, according to The Third International Consensus Definitions for Sepsis and Septic Shock (Sepsis-3) [7].

### ELISA analyses

BTLA in plasma was analysed in duplicate at 1:2 dilution with Human B- and T Lymphocyte Attenuator (BTLA) ELISA kit, Cusabio, detection range 0.45–30 ng/mL.

### Statistics

IBM SPSS Statistics for Windows, Version 25.0 (IBM corp. Armonk, NY, USA) was used for statistical calculations. Due to the high number of missing samples on admission, the focus of analyses was on measurements day 1–2 on. Missing data analysis was made, using chi-square, Fisher, unpaired t, or Mann-Whitney test, according to variable distribution, to compare

demographic and clinical characteristics in subjects with ≤1, or 2 or more missing plasma samples.

Non-normally distributed variables (sBTLA, CRP, and mHLA-DR) were log-transformed in further analyses. The sBTLA concentrations referred to in the results section are $\log_{10}$ sBTLA.

Cox regression was used to assess time to 90-day and 1-year mortality, in relation to sBTLA on day 1–2 and day 7, with complete follow up for all patients. sBTLA was evaluated on a continuous scale, and as a binary marker (high vs. low), where the threshold concentration was defined as the nearest even concentration above the highest concentration in the control group. Adjustments were made for age, sex, comorbidity on a categorical scale (0, 1–2, ≥3), and severity (day 1 Δ SOFA), on a continuous scale.

We used logistic regression to estimate the ability of sBTLA, on day 1–2, to differentiate sepsis from non-septic BSI, and on day 1–2 and 7 to predict survival on day 90 and 1 year, by calculating the area under the Receiver Operating Characteristic (ROC) curve (AUC) of the sensitivity and 1-specificity.

Linear Mixed Model with random intercept, and sBTLA as outcome, was used to evaluate sBTLA dynamics in patients over days 0–28, as well as to study age covariate-adjusted associations with inflammatory and immunosuppression markers from day 1–2 to 28, followed by stratified analyses, at individual time-points. Days after hospital admission was a fixed factor. Association coefficients are presented as mean difference (in text), and B (in figures), with 95% CI.

Unadjusted and age-adjusted linear regression with sBTLA as outcome was used to assess differences in sBTLA in patients and controls on day 0 and 28 and on day 1–2 to assess differences depending on day 1 ΔSOFA categorized as ≤3 and ≥4.

Patients were classified according to bacterial etiology: 1) *E. coli*, 2) *S. aureus*, 3) *S. pneumoniae*, and 4) other. Linear Mixed Model with random intercept, and sBTLA as outcome, was used to evaluate age-adjusted mean differences of sBTLA over days 1–2 to 28, between etiology groups 1–3. Fixed factors were days after hospital admission, etiology, and their interaction term. Autoregressive heterogeneous correlation structure was selected from best fit regarding Akaike information criteria. P-values comparing etiology groups were Bonferroni-corrected.

Statistical significance was defined as $p < 0.05$.

## Results

### Study population

Altogether, 116 patients were enrolled. Eight were excluded due to growth of non-pathogenic bacteria (n = 5), or no available plasma samples (n = 3), leaving 108 valid subjects. Plasma was available for sBTLA analysis from 46 patients on admission, 96 patients on day 1–2, 70 patients on day 3, 85 patients on day 7, 77 patients on day 14, and 70 patients on day 28. Subjects with 2 or more missing plasma samples (n = 60) were older (median age 73 vs. 69, p = 0.03) but not statistically significantly differing with regard to other baseline characteristics, bacterial etiology, disease severity, or short or long term mortality from those with ≤1 missing samples (n = 48), S1 Table.

Patient characteristics and control demographics are detailed in Table 1. The main sources of infection were pneumonia, urinary tract infection, skin/soft tissue, and joint infection. Fifty-four subjects (50%) met criteria for sepsis. Fourteen (13%) were treated in the ICU at some point during their hospital stay, due to need of organ supportive therapy, such as vasoactive drugs, or mechanical ventilation, as a result of sepsis. Ten out of 108 (9.3%) patients were immunocompromised at baseline, as defined by APACHE II criteria [28]. The frequency of

**Table 1. Patient and control characteristics.**

| | BSI (N = 108) | Controls (N = 31) |
|---|---|---|
| **Demographics** | | |
| Age, mean years (SD) range | 70 (14.6) 24–93 | 53 (47–59) |
| Sex, female | 49 (45) | 7 (23) |
| **Comorbidities** | | |
| No comorbidities | 37 (34) | |
| Ischemic heart disease | 21 (19) | |
| Congestive heart failure | 16 (15) | |
| Peripheral vascular disease | 5 (5) | |
| Diabetes with complications | 14 (13) | |
| Chronic lung disease | 8 (7) | |
| Connective tissue disease | 9 (8) | |
| Moderate/severe kidney disease | 7 (6) | |
| Chronic liver disease | 0 | |
| Malignancy the last 5 years | 5 (5) | |
| Immunosuppression | 10 (9) | |
| **Day 1 Δ-SOFA[1]** | | |
| 0–1 | 54 (50) | |
| 2 | 22 (20) | |
| 3–4 | 19 (18) | |
| ≥5 | 13 (12) | |
| **Pathogen** | | |
| *E. coli* | 25 (23) | |
| *S. aureus* | 27 (25) | |
| *S. pneumoniae* | 29 (27) | |
| Beta-hemolytic streptococci | 7 (6) | |
| Other gram negatives | 11 (10) | |
| Other gram-positives | 5 (5) | |
| Polymicrobial | 4 (4) | |
| Site of infection | | |
| Lungs/airways | 29 (27) | |
| Urinary tract | 26 (24) | |
| Abdominal | 7 (7) | |
| Skin/soft tissue | 10 (9) | |
| Joint/bone | 11 (10) | |
| Endocarditis | 8 (7) | |
| Primary BSI | 2 (2) | |
| Graft infection | 2 (2) | |
| Meningitis | 3 (3) | |
| Unknown | 10 (9) | |

Data in presented as N (%), unless otherwise stated.

[1]The increase from the patients baseline SOFA score.

antibiotic resistance was low: one subject had growth of ESBL-producing *E. coli*, and none had MRSA. The controls had a mean age of 53 years (SD 8.3, range 41–73) and 7 (23%) were female.

## sBTLA in BSI and controls

In patients, sBTLA was statistically significantly negatively associated with age on day 1–2, a decrease of -0.004 (95% CI 0.009 to 0.0, p = 0.045) $\log_{10}$ sBTLA units per year, but not at other time-points. In controls, there was no significant association between sBTLA and age.

Mean $\log_{10}$ sBTLA concentrations were statistically significantly higher in subjects with bacteremia, compared to controls. On day zero, unadjusted mean difference 0.43, 95% CI 0.24–0.62, p<0.001, and age-adjusted mean difference 0.47 (95% CI 0.25–0.70) p<0.001. On day 28, unadjusted mean difference 0.14 (95% CI 0.04–0.25) p<0.008, age-adjusted mean difference 0.13 (95% CI 0.01–0.25) p = 0.033, Fig 1. The median peak day of sBTLA was day 1–2. The linear mixed model showed that mean sBTLA decreased by 34.1% (95% CI 13.7 to 49.6) from admission to day 28 in subjects with BSI, with a mean decrease by 7.6% (95% CI 4.7 to 10.5) per time point.

## Disease severity

The association between day 1–2 sBTLA and day 1 ΔSOFA was non-linear, Fig 2. High ΔSOFA-scores ($\geq$4, n = 20) were associated with higher mean $\log_{10}$ sBTLA than low scores ($\leq$3, n = 76), mean difference 0.35 (95% CI 0.22–0.48, p<0.001. However, sBTLA did not perform well in differentiating sepsis (ΔSOFA score $\geq$2) from non-septic BSI at this time-point, AUC 0.54 (95% CI 0.43–0.66).

## Mortality

Among all 108 study patients, death from any cause was 4.6% (n = 5) on day 28, 11.1% (n = 12) on day 90, and 20.4% (n = 22) at one year. 7 patients died as a result of uncontrolled infection during hospital care for the index BSI, out of which 3 were not admitted to intensive care due to treatment limitations.

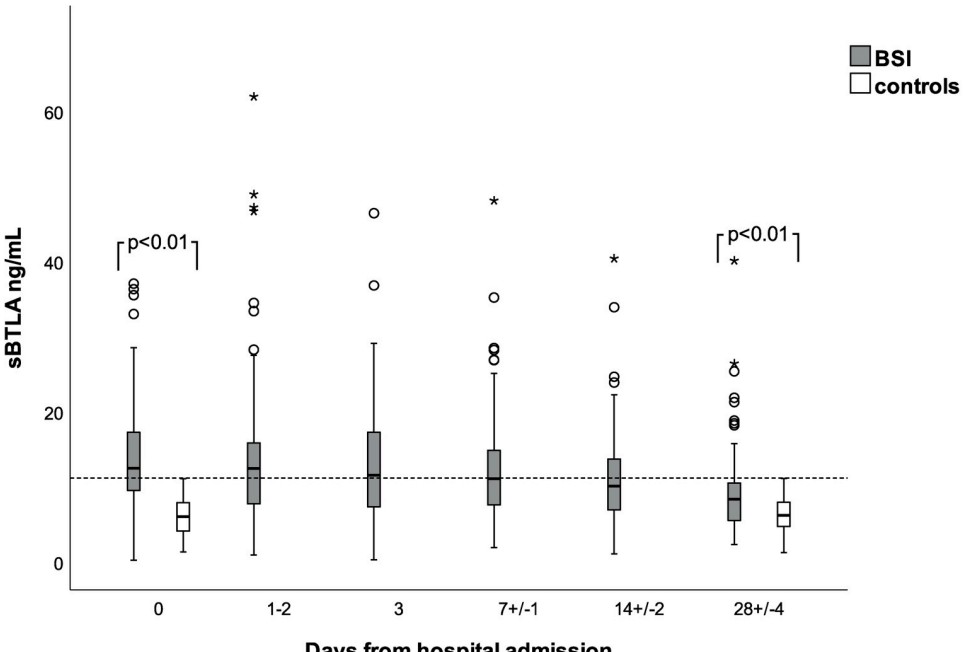

**Fig 1. sBTLA concentrations in BSI and controls from baseline to day 28.** Dotted line at 11 ng/mL. Outliers >80 ng/mL have been excluded from the figure to increase clarity (day 0: n = 3, day 3: n = 1, day 7 n = 1, day 14: n = 1).

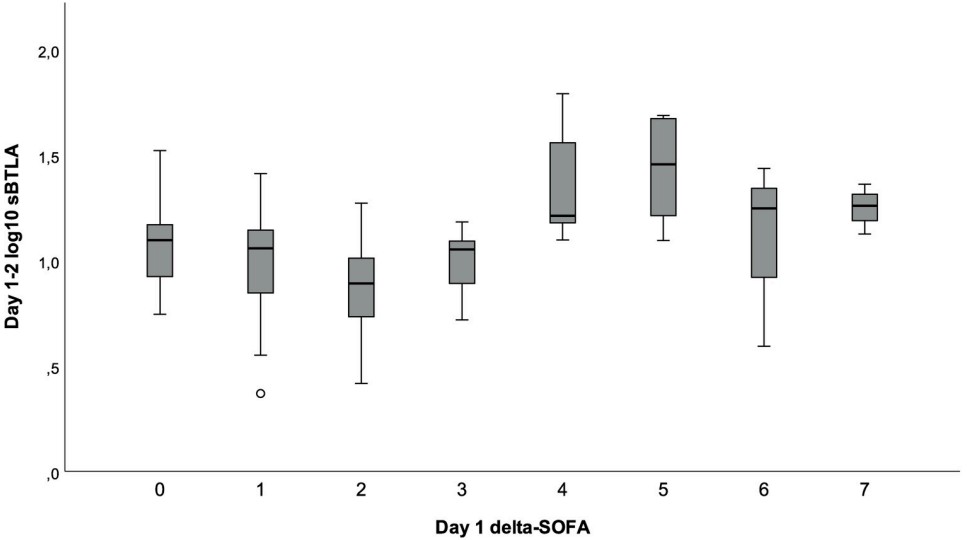

**Fig 2. Day 1–2 sBTLA concentrations and day 1 delta-SOFA.** ΔSOFA: 0 (n = 20), 1 (n = 27), 2 (n = 18), 3 (n = 11), 4 (n = 7), 5 (n = 6), 6 (n = 3), and 7 (n = 4).

On day 1–2, unadjusted HR, for time to 90-day mortality, was 4.6 (95% CI 0.6–34) p = 0.138, per log sBTLA unit, and adjusted (age, sex, ΔSOFA and comorbidity) HR was 114 (3.8–3409) p = 0.006. Unadjusted HR for sBTLA ≥11 (compared to <11) was 2.2 (95% CI 0.6 to 8.1) p = 0.236, and adjusted HR was 5.7 (95% CI 1.1–29) p = 0.034, Table 2.

On day 1–2, unadjusted HR, for time to 1-year mortality, was 2.3 (95% CI 0.5–11) p = 0.300 per log sBTLA unit, and adjusted HR was 10 (0.9–111) p = 0.061. Unadjusted HR for sBTLA≥11 (compared to <11) was 2.3 (95% CI 0.8 to 6.2) p = 0.114, and adjusted HR was 4.3 (95% CI 1.3–14) p = 0.016, Table 2.

On day 7, unadjusted HR, for time to 90-day mortality, was 5.4 (95% CI 0.7–40) p = 0.098 per log sBTLA unit, and adjusted HR was 22 (1.3–371) p = 0.034. Unadjusted HR for sBTLA ≥11 (compared to <11) was 9.7 (95% CI 1.2 to 77) p = 0.031, and adjusted HR was 17 (95% CI 1.8–160) p = 0.013, Table 2 and S1 Fig. AUC for day 7 sBTLA to predict mortality at 90 days was 0.72 (95% CI 0.59–0.84), Fig 3A.

**Table 2. Relative risk for 90-day and 1-year mortality.**

| | | | | 90-day mortality | | | | | | 1-year mortality | | | |
|---|---|---|---|---|---|---|---|---|---|---|---|---|---|
| | BSI n = 96 | Events[1] | Rate[2] | Unadjusted | | Adjusted[3] | | Events[1] | Rate[2] | Unadjusted | | Adjusted[3] | |
| | | | | HR (95% CI) | p | HR (95% CI) | p | | | HR (95% CI) | p | HR (95% CI) | p |
| **Day 1–2** | | | | | | | | | | | | | |
| log10 sBTLA, per unit | 96 | 12 | | 4.6 (0.6–34) | 0.14 | 114 (3.8–3409) | <0.01 | 20 | | 2.3 (0.5–11) | 0.30 | 10 (0.9–111) | 0.06 |
| sBTLA<11 | 40 | 3 | 32 | Ref | | Ref | | 5 | 14 | Ref | | Ref | |
| sBTLA≥11 | 56 | 9 | 70 | 2.2 (0.6–8.1) | 0.24 | 5.7 (1.1–29) | 0.03 | 15 | 33 | 2.3 (0.8–6.2) | 0.11 | 4.3 (1.3–14) | 0.02 |
| | BSI n = 85 | Events[1] | Rate[2] | Unadjusted | | Adjusted[3] | | Events | Rate[2] | Unadjusted | | Adjusted[3] | |
| **Day 7** | | | | HR (95% CI) | p | HR (95% CI) | p | | | HR (95% CI) | p | HR (95% CI) | p |
| log10 sBTLA, per unit | 85 | 10 | | 5.4 (0.7–40) | 0.10 | 22 (1.3–371) | 0.03 | 17 | | 6.1 (1.4–30) | 0.02 | 24 (2.8–208) | <0.01 |
| sBTLA<11 | 42 | 1 | 11 | Ref | | Ref | | 2 | 5 | Ref | | Ref | |
| sBTLA≥11 | 43 | 9 | 102 | 9.7 (1.2–77) | 0.03 | 17 (1.8–160) | 0.01 | 15 | 47 | 8.8 (2.0–39) | <0.01 | 15 (2.8–76) | <0.01 |

[1] Number of deaths.

[2] Mortality rate defined as deaths per 100 person years at risk.

[3] For age, sex, delta-SOFA, and comorbidity (Charlson score (0, 1–2, ≥3).

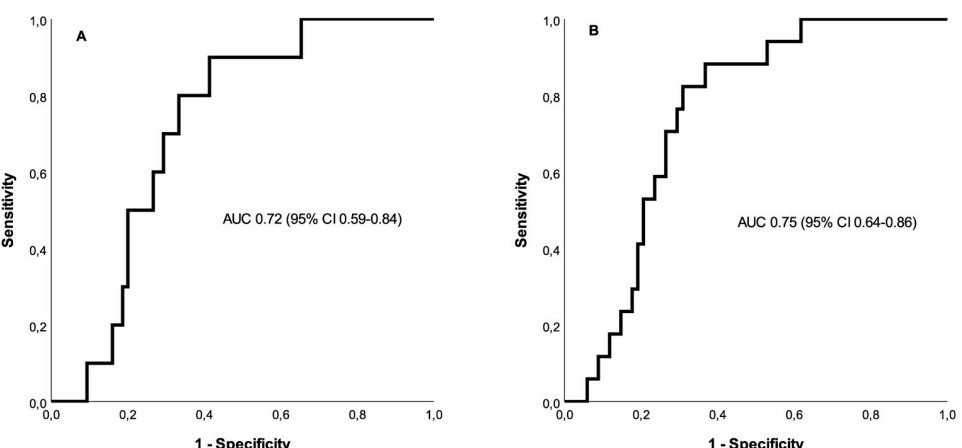

**Fig 3. Prognostic capacity of sBTLA on day 7.** (A) ROC for sBTLA on day 7 to predict 90-day mortality (B) ROC for sBTLA on day 7 to predict 1-year mortality.

On day 7, unadjusted HR, for time to 1-year mortality, was 6.1 (95% CI 1.4–30) p = 0.018 = per log sBTLA unit, and adjusted HR was 24 (2.8–208) p = 0.004. Unadjusted HR for sBTLA ≥11 (compared to <11) was 8.8 (95% CI 2.0–39) p<0.004, and adjusted HR was 15 (95% CI 2.8–76) p = 0.001, Table 2 and Fig 4. AUC for day 7 sBTLA to predict mortality at one year, was 0.75 (95% CI 0.64–0.86), Fig 3B.

Significant associations between day 1–2 and day 7 sBTLA and mortality at 90 days and 1 year remained after adjustment for the presence of immunosuppression at baseline, S2 Table.

## Associations with markers of inflammation and immunosuppression

The linear mixed model analysis, with sBTLA as outcome, and adjustment for time, showed a significant positive association with CRP, on average an increase of 0.15 (95% CI 0.09 to 0.20)

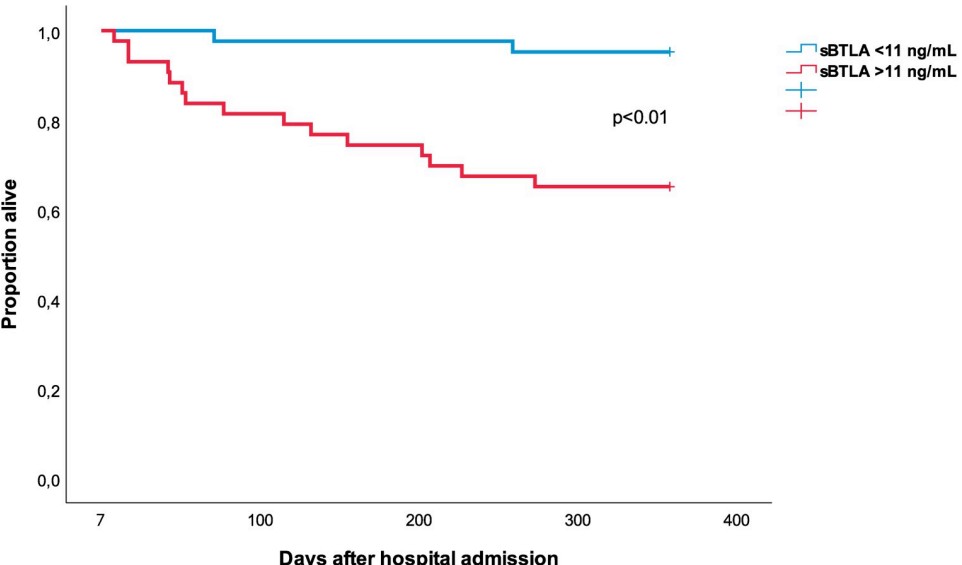

**Fig 4. Survival curves for high and low sBTLA on day 7.** sBTLA <11 ng/mL (n = 42), and sBTLA ≥11 ng/mL (n = 43). The p-value shown in the figure is after adjustment for age, sex, day 1 ΔSOFA, and comorbidity.

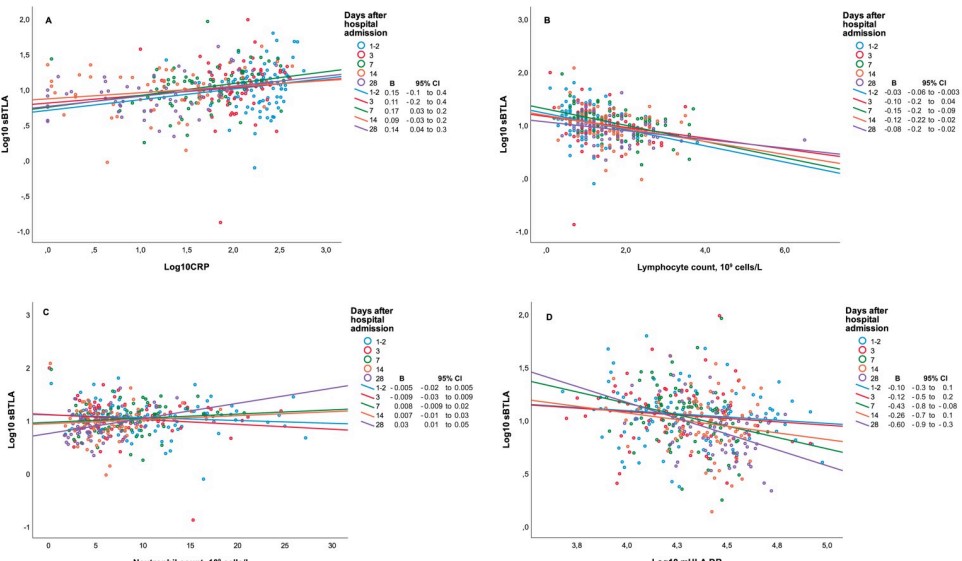

**Fig 5. Associations between sBTLA concentrations and immunological markers.** There was an overall significant association with CRP (A), and an overall negative association with lymphocyte count (B). There was no statistically significant association with neutrophil count (C), or with mHLA-DR (D). Association coefficients, and 95% CI (in parentheses), are shown for each individual time-point.

p<0.001 sBTLA log units per log CRP increase, and a significant negative association with lymphocyte count, i.e. a decrease of -0.03 (95% CI -0.05 to -0.008) p<0.008 sBTLA log units per lymphocyte count increase. There was no significant association with neutrophil count (p = 0.876), or mHLA-DR (p = 0.314), Fig 5. Associations at individual time-points are presented in the figure. In addition to the above, there was no association between sBTLA and neutrophil to lymphocyte count ratio (p = 0.26).

## Bacterial etiology

The linear mixed model analyses showed no statistically significant interaction effect, that is, no significant overall difference in mean BTLA dynamics over time between the etiology groups (p = 0.41). When the etiology groups were compared at different time-points, no major differences in sBTLA concentrations were observed between *S. aureus*, *E. coli* and *S. pneumoniae* etiology, Fig 6.

## Discussion

This study assessed sBTLA over 28 days after admission for bloodstream infection. The main finding is that sBTLA is associated with increased mortality at 90 days and 1 year. Additionally, we show for the first time that sBTLA is negatively associated with lymphocyte count.

We find that high sBTLA concentrations measured on day 1–2 and 7 days after hospital admission are associated with an increased risk of death before 90 days and one year. The association is independent of disease severity, age, and baseline comorbidities, and is most notable for failure to normalize sBTLA after 1 week's hospitalization. We previously demonstrated an association with early mortality, for high sBTLA concentration measured ≤24 hours after a diagnosis of severe sepsis or septic shock, in ICU-treated patients. This association was stronger in time-varying analysis, also indicating that the trajectory of sBTLA concentrations is of importance [22]. While one third of deaths in the studied cohort could be directly related to

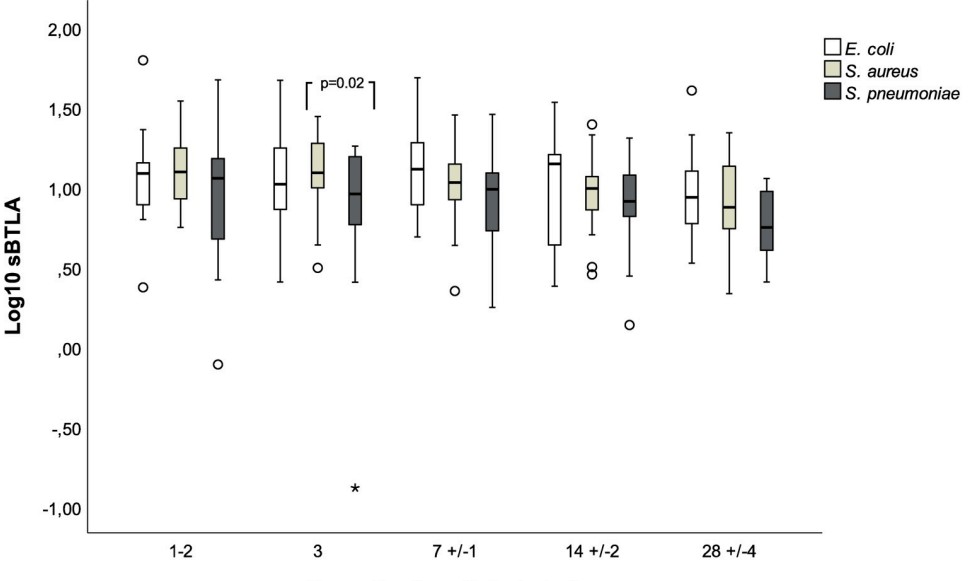

**Fig 6. BTLA concentrations in main etiology groups day 1–2 to day 28 post hospital admission.** Day 1–2: *E. coli* N = 20, *S. aureus* N = 24, *S. pneumoniae* N = 28 (*E. coli* vs *S. aureus* p = 1.0, *E. coli* vs *S. pneumonia* p = 0.4, *S. aureus* vs *S. pneumonia* p = 0.2). Day 3 *E. coli* N = 18, *S. aureus* N = 20, *S. pneumoniae* N = 17 (*E. coli* vs *S. aureus* p = 1.0, *E. coli* vs *S. pneumonia* p = 0.1). Day 7: *E. coli* N = 17, *S. aureus* N = 24, *S. pneumoniae* N = 25 (*E. coli* vs *S. aureus* p = 1.0, *E. coli* vs *S. pneumonia* p = 0.1, *S. aureus* vs *S. pneumonia* p = 0.2). Day 14: *E. coli* N = 14, *S. aureus* N = 19, *S. pneumoniae* N = 23 (*E. coli* vs *S. aureus* p = 1.0, *E. coli* vs *S. pneumonia* p = 0.9, *S. aureus* vs *S. pneumonia* p = 1.0). Day 28: *E. coli* N = 15, *S. aureus* N = 16, *S. pneumoniae* N = 21 (*E. coli* vs *S. aureus* p = 0.7, *E. coli* vs *S. pneumonia* p = 0.2, *S. aureus* vs *S. pneumonia* p = 1.0).

failure to clear the infection associated with the BSI, we did not have information on the cause of death in the majority of cases. The association between sBTLA and mortality may be due to immunological changes that occurred as a result from the index BSI. However, it could also reflect a pre-existing (i. e. before the infection) immune pathology that is associated with premature death. Experimental sepsis models have shown that BTLA deficient mice have a survival benefit, but treating wild type mice with anti-BTLA monoclonal antibody is unfavorable [29, 30]. It is possible that interference by sBTLA can account for these conflicting results. A recent study on circulating levels of 14 immune-checkpoint related proteins in patients with clear cell renal cancer, showed that high levels of sBTLA alone, were associated with an increased death risk [25]. Soluble BTLA has also been shown to have prognostic significance in other solid malignancies [24, 26, 31].

This study finds the strongest association with mortality for the failure to normalize sBTLA by day 7. Recent studies support dynamic assessment of inflammation and immunosuppression markers. In a retrospective study of patients with bacteremia and sepsis, Drewry et al. found that lymphopenia on day 4, but not on admission, was associated with 28-day mortality [32]. Stortz et al. studied critically ill septic patients with different clinical trajectories. They compared absolute lymphocyte count and levels of mHLA-DR and sPD-L1, and found that there were no differences at early assessment points (< 4 days), but that those with protracted critical illness failed to return to normal levels from day 7 onwards [33]. In another study, patients were followed until one year after sepsis, and the authors identified a patient phenotype with sustained inflammation and immunosuppression (elevated C-reactive protein (CRP) and sPD-L1), that was associated with increased long-term mortality and risk of hospital readmission [5].

This study shows an association between sBTLA and decreased lymphocyte count, which might indicate that sBTLA is linked to the process of immunosuppression in sepsis. There was however no significant overall association with mHLA-DR, another established marker of an immunosuppressed phenotype in sepsis [34]. We also find that sBTLA concentrations are positively associated with CRP, i e. with ongoing inflammation. In a previous study, we found no correlation between sBTLA and CRP, measured early in severe sepsis or septic shock. This discrepancy might be accounted for by the different analytic methods, where we in the present study took into account all measurements of sBTLA and CRP, which increases statistical power. Our findings support the latest model describing sepsis pahophyisiology beyond the acute phase, i e. the Persistent Inflammation, Immunosuppression, and Catabolism Syndrome (PICS), in which sustained elevation of CRP along with lymphopenia and low serum albumin is associated with poor outcome, and suggest that sBTLA might be an interesting addition to these established markers for immune monitoring [35].

We find that sBTLA concentrations peak early in BSI, decrease gradually, but on a group level remain elevated compared to controls 28 days post hospital admission. We do not find an overall age association for sBTLA in patients with BSI or controls. sBTLA was previously shown to exist at low concentrations, but increasing with age, in serum from healthy subjects [36]. A recent study by Gorgulho et al. showed increased levels of sBTLA in patients with advanced cancer compared to healthy controls, but no significant difference in in older (>67) compared to younger patients [31]. We previously reported that sBTLA is higher in sepsis and other critical illnesses, compared with healthy controls [22]. Monaghan et al. recently made similar findings, in human sepsis, as well as in a murine sepsis model, and they also showed that sBTLA is an alternative mRNA splice product, that is produced to a higher degree relative to the full length receptor, in critical illness in mice [23]. As to studies on cell-bound BTLA expression, results are conflicting, ranging from higher to lower CD4+ BTLA expression in sepsis compared to healthy volunteers [13, 37, 38]. The target cells and immunoregulatory effect of sBTLA remains to be fully clarified. A study by Han et al. showed that sBTLA can block BTLA-HVEM interaction, and in an anti-tumor vaccine model, they found that co-treatment of splenocytes with sBTLA resulted in increased IL-2 and IFN-γ and T cell cytotoxic activity [39]. Similarly, the Monaghan study also demonstrated that sBTLA increased splenocyte proliferation in vitro [23].

Among BSI patients, of which 50% fulfilled sepsis criteria, we find no linear association between sBTLA and severity, and that sBTLA cannot differentiate sepsis from non-septic BSI. sBTLA is however notably increased at higher ΔSOFA scores. In a previous study of ICU-treated patients with sepsis and septic shock, who had a mean enrolment SOFA of 10, we saw that sBTLA was correlated with SOFA score, and the strongest correlation was found at 24 hours after study enrolment [22]. A study by Shao et al. showed that BTLA expression on CD4 + cells is high in healthy controls compared to sepsis, with a gradual decrease of expression with increasing severity [38]. The alternative mRNA splice mechanism proposed by Monaghan et al., as described above, could account for an inverse relation between findings regarding soluble and membrane-associated BTLA in sepsis. On the other hand, Gorgulho et. al recently assessed the association between sBTLA and BTLA cell expression in a small number of cancer patients, and found a strong correlation between serum levels of sBTLA and the percentage of peripheral CD8+T cells expressing BTLA [31].

One limitation of this study is the size of the BSI and control cohorts, which were also unmatched with regard to age and comorbidities. Another, is the low number of samples at baseline, and the loss of sampling over time, which we tried to account for with missing samples analyses. The number of deaths at 90 days and 1 year is low, resulting in wide confidence intervals of hazard ratio estimates. An evaluation of the prognostic capacity of sBTLA in larger

studies is therefore warranted. Furthermore, the association we report is with all-cause mortality. Finally, we chose a cut-off for sBTLA-concentration that was based on the upper limit concentration in the control group, and it is possible that the normal range in different populations differs from that in this sample. For reference, there is however one study that supports this choice of cut-off as the above "normal" [36].

Strengths of this study include the clear etiologic definition of the study cohort. There was a low prevalence of antibiotic resistance, thus treatment failure due to ineffective antibiotics is less likely to have affected the outcome. The range of severity from uncomplicated bloodstream infection to septic shock also reflects the protean presentations of bacterial infections.

It has been suggested that future adjuvant sepsis strategies must be individualized, which requires readily available clinical and biomarker-guided phenotyping. This was recently addressed in two retrospective cohort studies [40, 41]. The present study further supports that sBTLA may hold potential as a marker of disturbed immune homeostasis, and a prognostic indicator in sepsis. Together with the increasing number of reports promoting BTLA as a prognostic marker in cancer, mechanistic studies on how, and to what extent sBTLA modifies the immune response are motivated. This can also shed light on whether the BTLA-HVEM pathway might be a target for immune intervention. It is however possible that the observed association between increased levels of sBTLA and mortality reflects an immunological vulnerability, or anergy, in some individuals, which increases the risk of a life-threatening infection. To address that question, we need studies on sBTLA dynamics with relevant control groups, i e. non-infected individuals with increased risk for BSI and sepsis, and we need longer follow-up times with documentation of the cause of death, to better understand the association with mortality. The utility of sBTLA, alone, or in combination with for example CRP and lymphocyte count, in immune profile assessment, must also be validated in larger studies on septic and other dysregulated immunological conditions.

## Conclusions

We show that soluble B and T lymphocyte attenuator increases in community-acquired bloodstream infection and remains elevated, on a group level, compared with healthy controls up to 28 days. Sustained elevation of sBTLA one week after hospital admission is notably associated with an increased risk of 90-day and 1-year mortality, and sBTLA is associated with decreased lymphocyte count. This suggests that sBTLA might be a useful indicator of sustained immune dysregulation, and a prognostic tool in sepsis.

## Supporting information

**S1 Table. Missing sample analysis.** Baseline and clinical characteristics in patients with ≤1 and ≥2 missing samples.
(PDF)

**S2 Table. Survival analyses with adjustment for immunosuppression at baseline.**
(PDF)

**S1 Fig. sBTLA on day 7 in survivors and non-survivors at 90 days and 1 year.** (A) sBTLA on day 7 in patients alive at, or dead within 90 days post hospital admission (B) sBTLA on day 7 in patients alive at or dead within 1 year post hospital admission.
(PDF)

**S1 Data.**
(XLSX)

## Acknowledgments

We wish to thank the Department of Clinical Research Laboratory at Örebro University for laboratory work.

## Author Contributions

**Conceptualization:** Anna Lange, Kristoffer Strålin, Olof Hultgren.

**Data curation:** Anna Lange, Sara Cajander, Anders Magnuson, Kristoffer Strålin, Olof Hultgren.

**Formal analysis:** Anna Lange, Anders Magnuson.

**Funding acquisition:** Anna Lange, Sara Cajander, Kristoffer Strålin, Olof Hultgren.

**Investigation:** Anna Lange, Sara Cajander.

**Methodology:** Anna Lange, Anders Magnuson, Kristoffer Strålin, Olof Hultgren.

**Project administration:** Anna Lange, Sara Cajander, Kristoffer Strålin, Olof Hultgren.

**Software:** Anders Magnuson.

**Supervision:** Kristoffer Strålin, Olof Hultgren.

**Validation:** Olof Hultgren.

**Visualization:** Anna Lange, Anders Magnuson.

**Writing – original draft:** Anna Lange.

**Writing – review & editing:** Sara Cajander, Anders Magnuson, Kristoffer Strålin, Olof Hultgren.

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
