## [Decision Letter · Decision Letter 0]

11 Oct 2021

PONE-D-21-08617Sustained elevation of soluble B- and T- lymphocyte attenuator predicts long-term mortality in patients with bacteremia and sepsisPLOS ONE

Dear Dr. Lange,

Thank you for submitting your manuscript to PLOS ONE. After careful consideration, we feel that it has merit but does not fully meet PLOS ONE’s publication criteria as it currently stands. Therefore, we invite you to submit a revised version of the manuscript that addresses the points raised during the review process.

Please review comments made by reviewers and provide point by point response in your revised manuscript. Specifically please provide more information regarding the missing data.

We look forward to receiving your revised manuscript.

Kind regards,

Muhammad Adrish, MD, MBA, FCCP, FCCM

Academic Editor

PLOS ONE

Journal Requirements:

2. In your Methods section, please provide additional information about the participant recruitment method and the demographic details of your participants. Please ensure you have provided sufficient details to replicate the analyses such as: 

a) the recruitment date range (month and year), 

b) a statement as to whether your sample can be considered representative of a larger population, and 

c) a description of how participants were recruited.

Reviewers' comments:

Reviewer's Responses to Questions

**Comments to the Author**

1. Is the manuscript technically sound, and do the data support the conclusions?

Reviewer #1: Yes

Reviewer #2: Partly

Reviewer #3: Partly

2. Has the statistical analysis been performed appropriately and rigorously? 

Reviewer #1: Yes

Reviewer #2: Yes

Reviewer #3: I Don't Know

3. Have the authors made all data underlying the findings in their manuscript fully available?

Reviewer #1: Yes

Reviewer #2: Yes

Reviewer #3: Yes

4. Is the manuscript presented in an intelligible fashion and written in standard English?

Reviewer #1: Yes

Reviewer #2: Yes

Reviewer #3: Yes

5. Review Comments to the Author

Reviewer #1: This manuscript technically sound, and the data support the conclusions. The sample size seems acceptable .Data analysis was appropriate also it seems that all data are available. Various groups of patients with different types of infection was included in study , but the main limitation is the number of patients and it seems that we need more patients and also an study with a larger sample size,

Reviewer #2: Thank you for the invitation to review this manuscript.

The authors present data on serial sBTLA levels in a 2011-14 patient cohort admitted with acute illness due to confirmed bloodstream infection. Their principal finding was of an association between persistently elevated sBTLA levels at day 7 and mortality at 90 days and 12 months.

Comments

1. Could the authors please include a concise description of what is known about the biology of sBTLA? Are lymphocytes / specific lymphocyte subsets the origin? What differentiates cell surface bound from soluble BTLA? What is the fate and intravascular half-life of sBTLA? What effector cells does sBTLA act upon and to what effect? How is individual age known to affect the production and kinetics of sBTLA?

2. Could the authors please account for the apparent slow recruitment of 116 patients over 4 years and the 7 year gap between the close of recruitment and the submission of this paper?

3. Only 14 (13%) of patients were admitted to ICU during their acute hospital admission. Where any patients not admitted to ICU due to poor prognosis or for any other reason despite apparent indications for such an admission? What were the reasons for the ICU admission - specifically, how many were the consequence of the development of septic shock or acute severe organ(s) dysfunction due to the BSI that had precipitated hospital admission? How did ICU admission affect short and long term mortality risk?

4. I am surprised that the distribution of included patient ages is parametric, I would have expected it to be negatively skewed and reported as a median [IQR] - could the authors comment?

5. In Figure 1, the median value of sBTLA does appear to fall progressively over the 28 day period but by a very small increment compared to the range of measured values. If the authors analysed the delta sBTLA, and / or delta sBTLA as a percentage of the peak value, for each individual, does this provide a more insightful picture of the evolution of sBTLA levels? For example, percentage drop from peak value of both CRP and PCT are emerging as viable stopping criteria for antimicrobial therapy - hence the question.

6. The mortality data is fascinating, but requires more information. I presume that cause of death / death certification information is a matter of public record in Sweden? As such, could the authors please provide this data and factor it into their discussion. What does this data tell us about the apparent association between prolonged immune dysregulation and the pathology that causes apparent "premature" death? Or, does this data suggest that the index BSI itself occurred because of an occult pathology that would go on to cause death in the succeeding 12 months?

7. Given the association / correlation with widely used existing biomarkers, can the authors comment or speculate on the value of measuring sBTLA levels in addition to, or instead of CRP, for example. Furthermore, could the authors comment on the correlation between sBTLA levels and neutrophil : lymphocyte ratio, which is emerging as a more useful marker than absolute values of either?

8. Could the authors provide a figure that shows the differences in peak, day 7, and day 7 as a percentage of peak in sBTLA levels in survivors verses non-survivors at 28 days, 90 days and 12 months?

9. Could the authors please add more detail to their suggestions regarding what implications their findings should have in terms of the next questions / hypotheses research into sBTLA and other markers of immunosuppression should aim to address?

Reviewer #3: Dear Sirs,

I read both your papers with great interest.

Regarding the paper in review process: the new research actually expends your previous results, regarding sBTLA and sepsis/septic shock patients to the BSI patents. Were some of the patients in both studies? If so, please acknowledge that in the new study (same 31 patients in the control, or just a coincidence?)

I have some concerns regarding the fact that the statistics is too complex, and I think it should be validated by a statistician.

Row 93-95- in introduction do you state your previous research, or did you include a conclusion statement in introduction?

You have a lot of missing data, not only at admission time, but also during the data collecting process. How do you explain this fact?

Do you have a table with the demographic data of the controls? It would be interesting to compare their data with patients' data.

You have more than 100 patients in the patients' group and only 31 in the controls. Why did you decide do compare uneven groups?

6. PLOS authors have the option to publish the peer review history of their article (what does this mean?). If published, this will include your full peer review and any attached files.

Reviewer #1: **Yes: **Mohsen Rajaeinejad

Reviewer #2: **Yes: **Jonathan Ball

Reviewer #3: **Yes: **Oana Antal

---

## [Author Response · Author response to Decision Letter 0]

2 Dec 2021

See Response to reviewer (attached document).

---

## [Decision Letter · Decision Letter 1]

9 Mar 2022

Sustained elevation of soluble B- and T- lymphocyte attenuator predicts long-term mortality in patients with bacteremia and sepsis

PONE-D-21-08617R1

Dear Dr. Lange,

We’re pleased to inform you that your manuscript has been judged scientifically suitable for publication and will be formally accepted for publication once it meets all outstanding technical requirements.

Kind regards,

George Vousden

Deputy Editor-in-Chief

PLOS ONE

Additional Editor Comments (optional):

Reviewers' comments:

Reviewer's Responses to Questions

**Comments to the Author**

1. If the authors have adequately addressed your comments raised in a previous round of review and you feel that this manuscript is now acceptable for publication, you may indicate that here to bypass the “Comments to the Author” section, enter your conflict of interest statement in the “Confidential to Editor” section, and submit your "Accept" recommendation.

Reviewer #2: All comments have been addressed

Reviewer #3: All comments have been addressed

2. Is the manuscript technically sound, and do the data support the conclusions?

Reviewer #2: Yes

Reviewer #3: Yes

3. Has the statistical analysis been performed appropriately and rigorously? 

Reviewer #2: Yes

Reviewer #3: I Don't Know

4. Have the authors made all data underlying the findings in their manuscript fully available?

Reviewer #2: Yes

Reviewer #3: Yes

5. Is the manuscript presented in an intelligible fashion and written in standard English?

Reviewer #2: Yes

Reviewer #3: Yes

6. Review Comments to the Author

Reviewer #2: (No Response)

Reviewer #3: Thank you for replying to all my concerns. I have no further issues to raise, as you replayed to all I have asked for.

7. PLOS authors have the option to publish the peer review history of their article (what does this mean?). If published, this will include your full peer review and any attached files.

Reviewer #2: **Yes: **Jonathan Ball

Reviewer #3: No

---

## [Editor Report · Acceptance letter]

11 Mar 2022

PONE-D-21-08617R1 

Sustained elevation of soluble B- and T- lymphocyte attenuator predicts long-term mortality in patients with bacteremia and sepsis 

Dear Dr. Lange:

I'm pleased to inform you that your manuscript has been deemed suitable for publication in PLOS ONE. Congratulations! Your manuscript is now with our production department. 

Kind regards, 

on behalf of

Dr. George Vousden 

Staff Editor

PLOS ONE